# Mechanism of synergistic activation of Arp2/3 complex by cortactin and WASP-family proteins

Fred E. Fregoso [1,2], Malgorzata Boczkowska [1], Grzegorz Rebowski [1], Peter J. Carman[1,2], Trevor van Eeuwen [3] & Roberto Dominguez [1,2] ✉

Cortactin coactivates Arp2/3 complex synergistically with WASP-family nucleation-promoting factors (NPFs) and stabilizes branched networks by linking Arp2/3 complex to F-actin. It is poorly understood how cortactin performs these functions. We describe the 2.89 Å resolution cryo-EM structure of cortactin's N-terminal domain (Cort$_{1-76}$) bound to Arp2/3 complex. Cortactin binds Arp2/3 complex through an inverted Acidic domain (D20-V29), which targets the same site on Arp3 as the Acidic domain of NPFs but with opposite polarity. Sequences N- and C-terminal to cortactin's Acidic domain do not increase its affinity for Arp2/3 complex but contribute toward coactivation with NPFs. Coactivation further increases with NPF dimerization and for longer cortactin constructs with stronger binding to F-actin. The results suggest that cortactin contributes to Arp2/3 complex coactivation with NPFs in two ways, by helping recruit the complex to F-actin and by stabilizing the short-pitch (active) conformation, which are both byproducts of cortactin's core function in branch stabilization.

The cellular pool of actin is in constant flux between monomeric (G-actin) and filamentous (F-actin) states[1]. F-actin forms networks of different architectures[2]. A common architecture is that consisting of branched filament networks (dendritic networks), resulting from the filament nucleation and branching activities of Arp2/3 complex, a seven-subunit complex comprising two actin-related proteins (Arps: Arp2, Arp3) and five scaffolding proteins (ArpC1-ArpC5)[3]. Branched networks generate pushing forces that drive processes such as cell and organelle motility. Many proteins regulate Arp2/3 complex in these processes[3], including WASP-family nucleation promoting factors (NPFs), which activate Arp2/3 complex to nucleate the formation of a branch (daughter) filament on the side of a pre-existing (mother) filament[3–6]. NPFs help drive a conformational change that reorients the Arps from their position in the inactive complex, where they interact end-to-end[7], into an actin filament-like short-pitch conformation in the active complex at the branch junction[8–10]. Other Arp2/3 complex regulators include proteins that promote branch disassembly such as glial maturation factor (GMF)[11] and coronins[12], inhibitors such as Arpin[13], and the branch stabilizer cortactin[14].

Cortactin is found in association with branched networks throughout the cell, including in lamellipodia, endocytic/exocytic sites, invadosomes, and the comet tails of trafficking vesicles and pathogens[14–18]. Human cortactin is a 550 amino acid protein, comprising N-terminal acidic (NTA, residues 1-76), six and a half 37-amino acid cortactin repeats (CRs, residues 80-324), and C-terminal Src homology 3 (SH3, residues 492-550) domains. The NTA (referred to here as Cort$_{1-76}$) and CRs bind Arp2/3 complex and F-actin, respectively[16,18,19]. A long helix (Helical domain, residues 346-401) C-terminal to the CRs has been also implicated in F-actin binding[20]. Cortactin interacts with many cytoskeletal and signaling proteins, primarily through its SH3 domain[21], and is the target of Src-family

[1]Department of Physiology, Perelman School of Medicine, University of Pennsylvania, Philadelphia, PA, USA. [2] Biochemistry and Molecular Biophysics Graduate Group, Perelman School of Medicine, University of Pennsylvania, Philadelphia, PA, USA. [3] Laboratory of Cellular and Structural Biology, The Rockefeller University, New York, NY 10065, USA. ✉e-mail: droberto@pennmedicine.upenn.edu

tyrosine kinases that phosphorylate residues in a proline-rich region located between the Helical and SH3 domains[15,22].

Cortactin has been described as both an activator of Arp2/3 complex and a branch stabilizer[18,19,23–27]. However, cortactin differs from class-I (WASP-family) NPFs, both in structure and activity, and is thus classified as a class-II NPF. In classical WASP-family NPFs (hereafter referred to simply as NPFs), the region responsible for Arp2/3 complex activation is located at the C-terminus and comprises G-actin-binding WH2 domain(s) (W) and Arp2/3 complex-binding Central (C) and Acidic (A) domains[28]. Cortactin lacks W and C domains and its A domain is located at the N-terminus. Moreover, contrary to NPFs, cortactin binds F-actin and not G-actin and has low Arp2/3 complex activating activity[16,18]. Yet, several studies have noted strong synergy between cortactin and NPFs, which together activate Arp2/3 complex more potently than they do individually[18,19,23–25,29]. The cause of this synergy has been puzzling, since an early study using chemical cross-linking showed that cortactin and NPFs compete for binding to subunit Arp3 of Arp2/3 complex[19]. It later emerged that NPFs have two binding sites on Arp2/3 complex, a site on Arp2-ArpC1 (site 1) and a site on Arp3 (site 2)[6,30–33], suggesting that cortactin and NPFs could co-bind the complex by distinctly targeting these two sites. Three models of synergistic activation have been proposed: direct-activation, recruitment, and displacement models. According to the direct-activation model, cortactin shifts the equilibrium of Arp2/3 complex toward the active, short-pitch conformation of the Arps, somewhat analogous to classical NPFs[18,23]. The recruitment model holds that cortactin helps recruit NPF-primed Arp2/3 complex to the mother filament[19,24], a necessary step in the activation pathway[34–36]. Finally, the displacement model suggests that cortactin promotes the release of NPFs from nascent branch junctions, which is viewed as either a prerequisite for branch nucleation[25,29] or a way to accelerate the turnover of NPFs for successive cycles of nucleation[37]. Independent of which model is considered, cortactin remains bound after activation and stabilizes branch junctions, by linking Arp2/3 complex to either the mother[23,26,27] or daughter[38] filament. Here, we describe the cryo-electron microscopy (cryo-EM) structure of Cort$_{1-76}$ bound to Arp2/3 complex. Cortactin residues D20-V29 bind to a site on subunit Arp3 of Arp2/3 complex that coincides with one of the binding sites of the A domain of WASP-family NPFs, but with opposite polypeptide directionality. Structure-inspired biochemical studies suggest that sequences N- and C-terminal to the binding motif of cortactin contribute toward synergistic coactivation of Arp2/3 complex with NPFs by helping recruit the complex to F-actin and stabilizing the short-pitch conformation.

## Results

### Cryo-EM structure of Cort$_{1-76}$ bound to Arp2/3 complex

The cryo-EM structure of Cort$_{1-76}$ bound to Arp2/3 complex was determined at 2.89 Å resolution (Fig. 1, Table 1, Supplementary Movie 1 and Supplementary Figs. 1, 2). Previous work showed that mutations in the cortactin motif $^{20}$DDW$^{22}$ abolish binding to Arp2/3 complex[18,19]. This motif forms part of only ten cortactin residues visualized in the cryo-EM map ($^{20}$DDWETDPDFV$^{29}$). In the structure, Arp2/3 complex displays its most stable, inactive conformation, with the Arps interacting end-to-end, as observed in structures of Arp2/3 complex alone[7] or bound to regulatory proteins[6,39–41]. Cortactin binds at the interface between subdomains 3 and 4 of Arp3 (Fig. 1b, c). The binding site consists of a hydrophobic pocket, which hosts the side chain of cortactin residue W22, surrounded by positively charged amino acids (R230, K240, K244, K251, K254, R275, R329, R333, R334, R337) that create a favorable electrostatic environment for interactions with negatively charged amino acids of cortactin (D20, D21, E23, D25, D27). This site also binds the A domain of NPFs (NPF-binding site 2)[6,31] and the A domain of the inhibitor Arpin[41], and thus constitutes a hot spot for interactions of different Arp2/3 complex regulators. Curiously, however, cortactin binds this site with opposite polarity compared to

NPFs and Arpin (Fig. 1c). This finding is consistent with two observations. First, in cortactin the A domain is at the N-terminus whereas in NPFs and Arpin it is at the C-terminus. Second, the A domain of cortactin can be better aligned with those of NPFs and Arpin when inverted (Fig. 1c, middle). Therefore, cortactin interacts with Arp2/3 complex through an inverted A domain.

### Testing the direct-activation model of Arp2/3 complex by cortactin

During activation, Arp2/3 complex undergoes a conformational change that brings the Arps into a short-pitch conformation[8–10]. While studies have shown that the WCA region of WASP-family NPFs promotes this conformational change in solution[6,36,42], the structure of WCA-bound Arp2/3 complex displays the inactive conformation[6]. Therefore, as per the direct-activation model, we asked whether Cort$_{1-76}$ also shifted the equilibrium of Arp2/3 complex toward the short-pitch conformation in solution without this being reflected in the structure (Fig. 1b). To monitor the conformation of Arp2/3 complex, we used a short-pitch crosslinking assay developed for *S. cerevisiae* Arp2/3 complex[42] and adapted by us to human Arp2/3 complex[6]. In this assay, cysteine residues introduced by mutagenesis in Arp2 (L199C) and Arp3 (L117C) come within crosslinking distance by bismaleimidoethane (BMOE) only in the short-pitch conformation, such that quantification of the ~150 kDa Arp2-Arp3 crosslinking band in Western blots provides an estimate of the short-pitch transition (Fig. 2 and Supplementary Fig. 3). Under the experimental conditions used, crosslinking of Arp2/3 complex alone reached a plateau ~120 min after BMOE treatment. At plateau, BMOE likely separately reacts both cysteine residues of the remaining uncrosslinked fraction of Arp2/3 complex, which as a result can no longer form a short-pitch crosslink. Approximately 37% of Arp2/3 complex was crosslinked at plateau, consistent with the ability of the complex to transiently visit the short-pitch conformation without this resulting in activation[6]. The addition of N-WASP WCA increased the crosslinking rate 1.45-fold, resulting in a crosslinked fraction at plateau of 48%. In contrast, the addition of Cort$_{1-76}$, with or without N-WASP WCA, did not significantly change the crosslinking rate nor the crosslinked fraction. We conclude that, unlike class-I NPFs, cortactin does not shift the equilibrium of Arp2/3 complex toward the short-pitch conformation in solution, thereby ruling out the direct-activation model.

### Testing the recruitment model of Arp2/3 complex activation by cortactin

The recruitment model predicts that Arp2/3 complex coactivation with NPFs should increase for cortactin constructs with stronger binding to F-actin. To test this model, we studied four cortactin constructs, including Cort$_{1-76}$ and three constructs comprising different segments of the F-actin-binding CR and Helical domains (Fig. 3a, right and Supplementary Fig. 4). First, we analyzed the ability of these constructs to bind F-actin on their own. Consistent with previous findings[16,20], Cort$_{1-76}$ did not bind F-actin in high-speed cosedimentation experiments, whereas Cort$_{1-227}$, Cort$_{1-79/154-401}$, and Cort$_{1-401}$ cosedimented with F-actin to different extents (Supplementary Fig. 5). Generally, constructs with a longer F-actin-binding region cosedimented more abundantly: Cort$_{1-401}$ > Cort$_{1-79/154-401}$ > Cort$_{1-227}$. However, we do not report dissociation constants ($K_D$) for these interactions since cortactin not only binds but also bundles F-actin (Supplementary Fig. 6a)[29]. Moreover, in the absence of Arp2/3 complex, the interaction of cortactin with F-actin seems to lack specificity. Consistently, a 2.77 Å helical reconstruction of F-actin from the bundles formed with Cort$_{1-401}$ shows no extra density corresponding to Cort$_{1-401}$ (Supplementary Fig. 6). This suggests that the interaction of cortactin with F-actin is driven by non-specific electrostatic contacts; at neutral pH, F-actin has a net negative charge (pI of the actin

monomer = 5.23), whereas the CR region of cortactin has a pI of 8.9. The lack of cortactin density in the absence of Arp2/3 complex differs from a previous study that identified a recurring patch of extra density on F-actin that was attributed to cortactin[43]. The lower resolution of that study (~23 Å) and the use of negative staining EM probably explain the different results.

Next, we analyzed the ability of cortactin constructs to recruit Arp2/3 complex to F-actin. Full-length cortactin has been shown to increase the amount of Arp2/3 complex that cosediments with F-actin[26]. Here, we found that the amount of Arp2/3 complex cosedimenting with F-actin increased with all the cortactin constructs except Cort$_{1-76}$, and this increase was proportional to the ability of cortactin

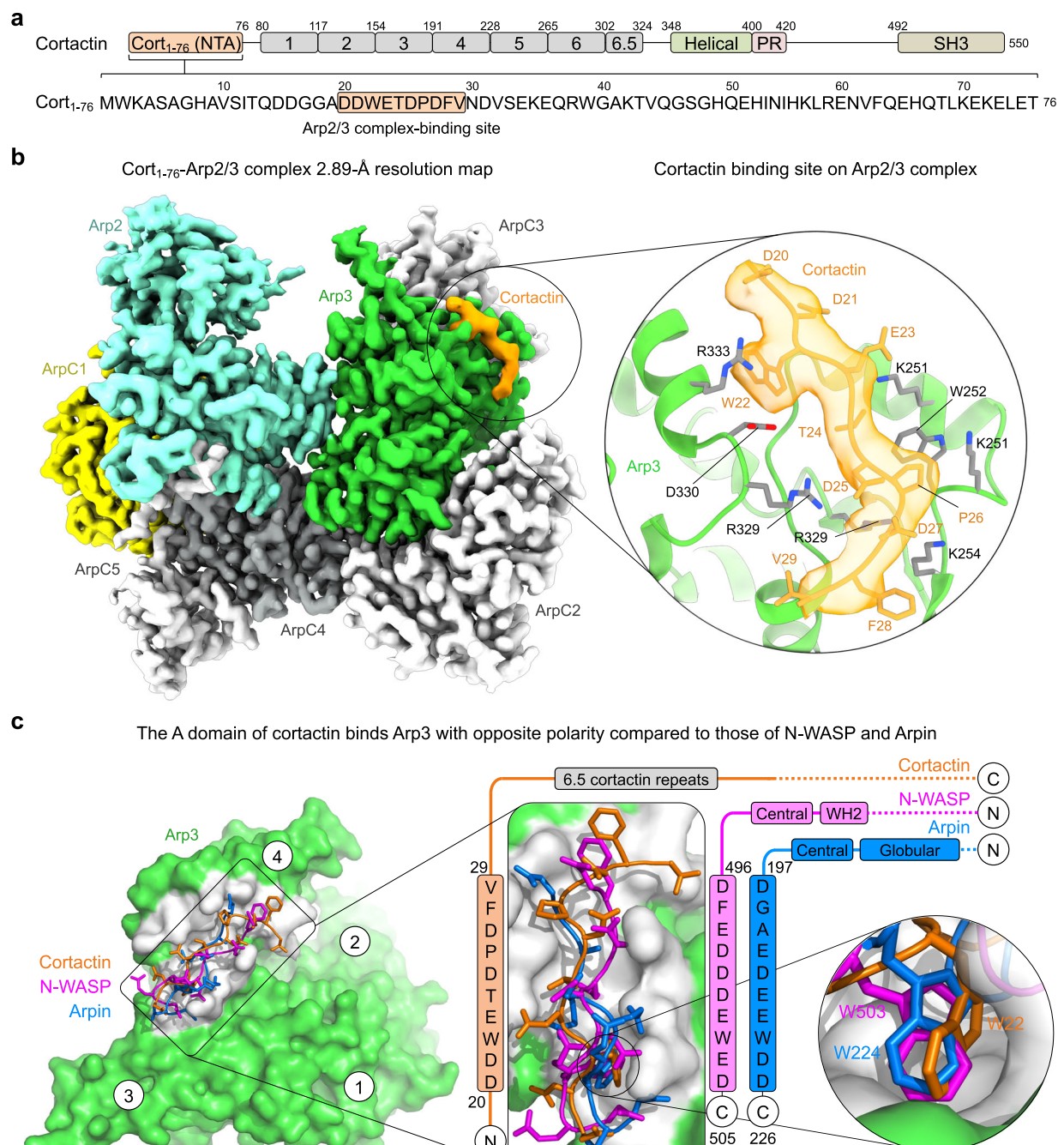

**Fig. 1 | Cryo-EM structure of the inverted acidic domain of cortactin bound to Arp2/3 complex. a** Domain diagram of human cortactin (UniProt: Q60598), highlighting the sequence of the fragment Cort$_{1-76}$ used in the cryo-EM structure and residues D20-V29 implicated in Arp2/3 complex binding. **b** Cryo-EM map of Cort$_{1-76}$ (orange) bound to Arp2/3 complex. Arp2/3 complex subunits are colored as follows: Arp2, cyan; Arp3, green; ArpC1, yellow; ArpC2, ArpC3, and ArpC5, light gray; ArpC4, dark gray. The inset shows cortactin residues D20-V29 and the corresponding cryo-EM map in orange, and the binding site on Arp3 (green cartoon

representation with gray side chains). See also Supplementary Fig. 2a. **c** Superimposition of the A domains of cortactin (orange), N-WASP (magenta, PDB code 6UHC), and Arpin (blue, PDB code 7JPN) on Arp3 (green and gray). Insets show the binding site (gray) of the A domains and the binding pocket of the conserved tryptophan of the three proteins. The sequences of the three A domains are shown on the side, vertically aligned according to the structures. Note that cortactin binds the site on Arp3 with inverted polarity compared to N-WASP and Arpin.

**Table 1 | Cryo-EM data collection, structure refinement, and validation**

| Data collection and processing | Cort$_{1-76}$ – Arp2/3 complex |
|---|---|
| Number of Micrographs | 4016 |
| Magnification | 105,000 |
| Voltage, keV | 300 |
| Total electron exposure, e$^-$/Å$^2$ | 41.6 |
| Defocus range, μm | −1.0 to −3.0 |
| Pixel size, Å | 0.42 |
| Symmetry | C1 |
| Initial no. particles | 1,461,919 |
| Final no. particles | 192,734 |
| Map resolution, Å | |
| FSC threshold 0.143 (0.5) | 2.89 (3.27) |
| Resolution range, Å | 2.7–4.8 |
| Refinement | |
| Initial models used (PDB codes) | 7JPN |
| Resolution refined structure, Å | 2.97 |
| FSC threshold | 0.5 |
| Map sharpening B factor, Å$^2$ | −89.87 |
| Model composition | |
| No. non-hydrogen atoms | 14,976 |
| No. residues | 1884 |
| No. ligands | ATP: 2, Mg: 2 |
| Correlation model vs. data | |
| CC (mask, box, peaks, volume) | 0.88, 0.80, 0.75, 0.87 |
| R.m.s. deviations | |
| Bond lengths, Å (no. > 4σ) | 0.005 (0) |
| Bond angles, ° (no. > 4σ) | 0.653 (1) |
| Validation | |
| MolProbity score | 1.77 |
| Clashscore | 2.28 |
| Rotamer outliers, % | 4.46 |
| Ramachandran plot | |
| Favored, % | 96.03 |
| Allowed, % | 3.97 |
| Disallowed, % | 0.00 |
| ADP (B-factors) | |
| Protein, Å$^2$ (min/max/mean) | 80.45/260.44/146.14 |
| Ligands, Å$^2$ (min/max/mean) | 89.11/171.90/144.98 |
| Accession codes | |
| EMDB | EMD-41135 |
| PDB | 8TAH |

constructs to bind F-actin on their own (Fig. 3a, left and Supplementary Fig. 7).

We then used the pyrene-actin polymerization assay to assess the ability of cortactin constructs to activate Arp2/3 complex with and without monomeric N-WASP WCA (Fig. 3b–e). These experiments led to several observations. First, Cort$_{1-401}$ and Cort$_{1-79/154-401}$, which more abundantly enhanced Arp2/3 complex cosedimentation with F-actin, activated Arp2/3 complex on their own, whereas Cort$_{1-76}$ and Cort$_{1-227}$ did not. Second, for a constant concentration of actin (2 μM), Arp2/3 complex (20 nM), and WCA (100 nM), the maximum polymerization rate first increased and then decreased with increasing concentrations of cortactin constructs. The increase in polymerization is expected for a synergistic activation mechanism of Arp2/3 complex by NPFs and cortactin. However, the decrease in polymerization at higher cortactin

concentrations is a new observation. At least in part, this effect is likely due to competition of cortactin constructs with Arp2/3 complex for binding to F-actin and increased filament bundling. However, this effect was also observed with Cort$_{1-76}$, which does not bind F-actin, suggesting that competition with WCA for binding to Arp3 also contributes to the decrease in polymerization at higher cortactin concentrations. Third, the magnitude of Arp2/3 complex coactivation with WCA correlated with the ability of cortactin constructs to bind F-actin (Supplementary Fig. 5b) and cosediment Arp2/3 complex (Fig. 3a). These results are all consistent with the recruitment model of Arp2/3 complex coactivation, whereby NPFs bind to site 1 and deliver actin to Arp2 whereas cortactin binds to Arp3 to help recruit the complex to F-actin. However, Cort$_{1-76}$, which did not recruit Arp2/3 complex to F-actin on its own, also synergized with monomeric WCA during Arp2/3 complex activation, albeit to a much lesser extent compared to longer cortactin constructs (Fig. 3e).

**Testing the displacement model of Arp2/3 complex activation by cortactin**

Strong support for the displacement model resulted from the observation that cortactin's NTA (a construct similar to Cort$_{1-76}$), which does not bind F-actin, synergizes with dimeric GST-WWCA during Arp2/3 complex activation[25]. The synergistic effect observed with GST-WWCA appeared much stronger than what we found above with monomeric WCA. To reproduce these results, we used here the same GST-WWCA construct used by these authors, which includes the two WH2 domains of N-WASP. It is important to note that, alone, WCA is a stronger activator of Arp2/3 complex than WWCA, both as monomeric[6,29,44] or dimeric[29] constructs. At a fixed concentration of Arp2/3 complex (20 nM) and GST-WWCA (2.5 nM), we observed a concentration-dependent increase in the polymerization rate with Cort$_{1-76}$ (0–30 μM) (Fig. 4a). The synergistic effect was indeed much stronger than with monomeric WCA (Fig. 3e). Moreover, a polymerization decay was not observed even at 6-fold higher concentration of Cort$_{1-76}$ compared to what was used with monomeric WCA. This is likely due to one arm of the GST-WWCA dimer being bound to Arp2-ArpC1 whereas the other arm, which is displaced from Arp3 by Cort$_{1-76}$, remaining close enough to deliver actin to Arp3.

We reasoned that if the displacement model were correct, direct competition with WCA for binding to Arp3 was all that was needed for synergistic activation. Therefore, we repeated this experiment using constructs Cort$_{1-33}$ and Cort$_{18-76}$, featuring deletions of the regions N- or C-terminal to the Arp2/3 complex-binding motif. Compared to Cort$_{1-76}$, the ability to coactivate Arp2/3 complex with GST-WWCA was dramatically reduced for Cort$_{18-76}$ (Fig. 4b) and eliminated for Cort$_{1-33}$ (Fig. 4c). By isothermal titration calorimetry, these three proteins bound Arp2/3 complex with ~ 1:1 stoichiometry and similar affinities (K$_D$ = 2.2–5.4 μM) (Fig. 4d). This result is consistent with the structure (Fig. 1), revealing a single cortactin-binding site on Arp3, mediated by residues D20-V29 present in all three proteins. Therefore, the three proteins should compete with NPF binding to Arp3 and, according to the displacement model, should have activated Arp2/3 complex synergistically with GST-WWCA, but only Cort$_{1-76}$ did. These findings have two important implications. First, the displacement model cannot explain the cooperative activation of Arp2/3 complex by cortactin and NPFs. Second, the highly conserved regions N- and C-terminal to the Arp2/3 complex-binding motif of cortactin (Supplementary Fig. 8) are also important for cooperative activation. These regions may help stabilize the active, short-pitch conformation and/or establish additional contacts between Arp2/3 complex and either the mother or daughter filament at the branch junction (Fig. 5). Similar results were obtained before[29], albeit these authors interpreted their data as supporting the displacement model (discussed below).

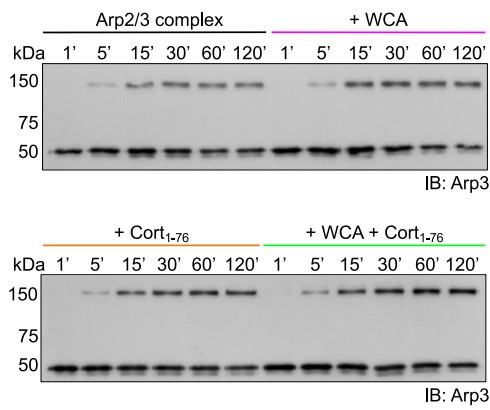

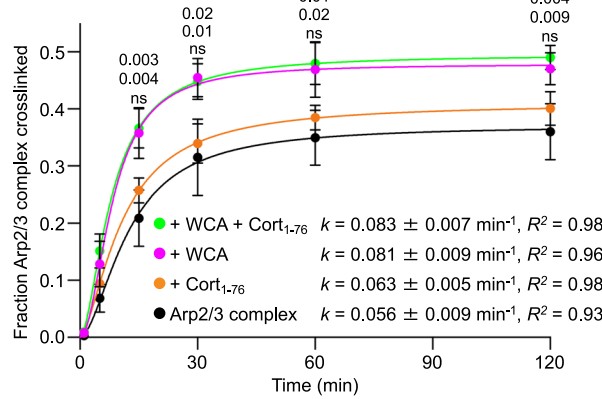

**Fig. 2 | Effect of Cort$_{1-76}$ and N-WASP WCA on the short-pitch transition of Arp2/3 complex.** Western blot analysis (Arp3 antibody, left) and densitometric quantification ($n=3$, right) of the cross-linked fraction of Arp2/3 complex alone (black) or with WCA (magenta), Cort$_{1-76}$ (orange), or WCA and Cort$_{1-76}$ (green). All the Western blots used for quantification are shown in Supplementary Fig. 3. Data were fit to a first-order exponential equation (kinetic parameters of the fit listed in the figure) and are reported as mean value ± standard deviation. The statistical significance of the measurements was calculated using an unpaired, two-tailed t-test ($P$-values listed in the figure). For each data point, $P$-values for pairwise comparisons with Arp2/3 complex alone (control) are shown from top-to-bottom for experiments with WCA + Cort$_{1-76}$, WCA, or Cort$_{1-76}$ (ns, non-significant). The source data are provided as a Source Data file.

## Discussion

The most stable conformation of Arp2/3 complex outside the branch is inactive, with the Arps interacting end-to-end, as observed in structures of the complex alone[7] and bound to GMF[39], SPIN90[40], N-WASP[6], and Arpin[41]. A different, active, conformation is observed at the branch junction[8–10], where the Arps interact side-by-side analogous to subunits of the short-pitch helix of F-actin. The structure of Cort$_{1-76}$-bound Arp2/3 complex displays the inactive conformation (Fig. 1b). When cortactin colocalizes with branched networks in cells[14–18] it is by interaction with the active conformation of Arp2/3 complex. However, when we analyze cortactin's ability to co-activate Arp2/3 complex with NPFs, we must also consider its interaction with inactive Arp2/3 complex. These two interactions likely differ somewhat from one another, since the complex changes conformation at the branch, but they are both important to understand cortactin's function. In the inactive complex, cortactin binds to a single site on Arp3, and the interaction is mediated by a conserved motif comprising a tryptophan and five negatively charged residues ($^{20}$DDWETDPDFV$^{29}$ (Supplementary Fig. 8). This sequence is similar to the A domains of NPFs and Arpin that bind the same site on Arp3[6,41], albeit with opposite polarity. Thus, cortactin binds Arp2/3 complex through an inverted A domain. The opposite polarity of cortactin at this site may have implications for its interaction with F-actin and synergy with NPFs during Arp2/3 complex activation (Fig. 5). It emerges that the same binding site on Arp3 is used by diverse Arp2/3 complex regulators, including activators, inhibitors, and branch stabilizers, and could be an excellent target for the design of drugs to control the activity of Arp2/3 complex.

Knowing how cortactin interacts with inactive Arp2/3 complex allowed us to design experiments to test three existing models of Arp2/3 complex coactivation by cortactin and NPFs: direct-activation, recruitment, and displacement models. Unlike WASP-family NPFs[6,36,42], cortactin did not shift the equilibrium of Arp2/3 complex toward the short-pitch conformation in solution (Fig. 2), allowing us to rule out the direct-activation model. In contrast, the data support the recruitment model, specifically the strong correlation observed in the ability of cortactin constructs to bind F-actin (Supplementary Fig. 5b), recruit Arp2/3 complex to F-actin (Fig. 3a), and coactivate Arp2/3 complex with NPFs (Fig. 3b–e). At higher concentrations, however, cortactin reduces NPF-mediated activation (Fig. 3b–e). This is likely due to three effects occurring in parallel: a) cortactin competes with Arp2/3 complex for binding to F-actin, b) Arp2/3 complex is excluded from the actin bundles formed by cortactin, and c) cortactin competes with NPF binding and actin delivery to Arp3. While our results provide strong support for the recruitment model, alone this model cannot explain some of our observations. Specifically, it cannot explain how the shorter construct Cort$_{1-76}$, which on its own: a) does not bind F-actin (Supplementary Fig. 5b), b) cannot recruit Arp2/3 complex to F-actin (Fig. 3a, c), and c) cannot stimulate the short-pitch transition (Fig. 2), is nevertheless capable of synergizing with monomeric (Fig. 3e) and dimeric (Fig. 4a) NPFs to activate Arp2/3 complex.

Can the displacement model explain the synergistic effect of Cort$_{1-76}$? The displacement model holds that cortactin synergizes with NPFs by accelerating their release from nascent branch junctions[25,29,37]. According to this model, any protein that competes with NPF binding to Arp3 at the branch should synergize with NPFs, but this is clearly not the case. Indeed, we found that constructs Cort$_{1-76}$, Cort$_{18-76}$, and Cort$_{1-33}$ all bind Arp2/3 complex with similar affinities in solution (Fig. 4d), but only Cort$_{1-76}$ effectively synergizes with GST-WWCA for Arp2/3 complex activation (Fig. 4a–c). Using slightly different constructs (cortactin 1-84, 15-79, 1-48, and 1–27), a previous study concluded that their data supported the displacement model[29]. Like us, these authors observed decreased synergy with every deletion of cortactin's N-terminal region. However, they concluded that the shorter constructs had weaker affinity for Arp2/3 complex at the branch junction (even if not for the isolated complex in solution), explaining why these constructs were less effective at displacing NPFs from nascent branch junctions. Although we lack direct evidence, it is indeed possible that Cort$_{1-76}$ has higher affinity for Arp2/3 complex at the branch junction, where the complex adopts a different conformation[8–10] (see below). But even if we assume this is true, Cort$_{18-76}$, and Cort$_{1-33}$ should still synergize at higher concentrations. Yet, at the highest concentration tested (30 µM), Cort$_{1-33}$ did not synergize, and Cort$_{18-76}$ enhanced the polymerization rate to the same level as 1–3 µM Cort$_{1-76}$, which represents a large discrepancy. Moreover, the synergistic effect of Cort$_{1-76}$ is the strongest when combined with dimeric GST-WWCA, which after being displaced from Arp3 can remain bound to the high affinity NPF-binding site on Arp2-ArpC1. These observations argue against the displacement model.

Based on what we now know about the structure of Arp2/3 complex at the branch[8–10] and bound to NPFs[6] and cortactin, we propose a different model of synergy for Cort$_{1-76}$. WASP-family NPFs bind to two sites on Arp2/3 complex, a site on Arp2-ArpC1 and a site on Arp3[6,30–33]. NPF binding to Arp2-ArpC1 and not Arp3 drives the equilibrium of Arp2/3 complex toward the short-pitch conformation[6]. However, NPF binding and actin delivery to Arp3 are crucial during activation, since a mutant that disrupts the A domain-binding site on Arp3 is inactive[6].

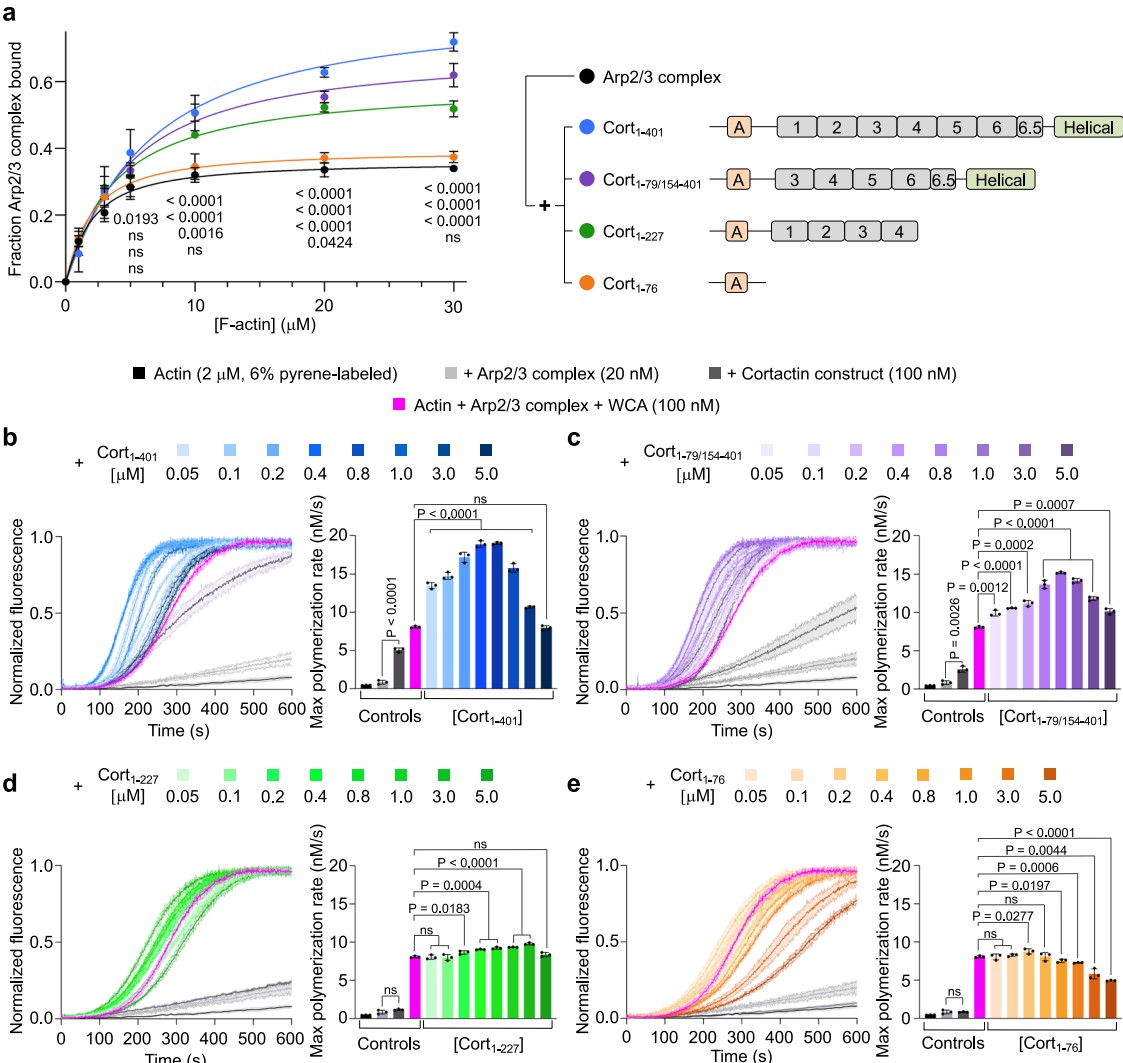

**Fig. 3 | Arp2/3 complex activation by cortactin constructs with or without monomeric N-WASP WCA. a** Densitometric quantification of cosedimentation experiments (left) of Arp2/3 complex with F-actin in the absence (black) or the presence of the cortactin constructs schematically shown on the right (color-coded). A characterization of the cortactin constructs by SDS-PAGE and the cosedimentation experiments used for quantification are shown in Supplementary Figs. 4, 7, respectively. The curves represent fits to a first-order exponential equation. For each data point, P-values for pairwise comparisons with Arp2/3 complex alone (control) are shown from top-to-bottom for experiments with Cort$_{1-401}$, Cort$_{1-79/154-401}$, Cort$_{1-227}$, or Cort$_{1-76}$ (ns, non-significant). **b–e** Left, Time-course of actin polymerization under the following conditions: actin alone (black), actin +

Arp2/3 complex (light gray), actin + Arp2/3 complex + indicated cortactin constructs (dark gray), actin + Arp2/3 complex + monomeric N-WASP WCA (magenta), actin + Arp2/3 complex + WCA + indicated cortactin constructs at increasing concentrations (color coded as in part **a**, with darker shades representing higher concentrations as indicated). Source Data are provided as a Source Data file. Data are shown as the average curve from three independent experiments with s.d. error bars in lighter color. Right, Maximum polymerization rates calculated from the polymerization curves shown on the left ($n = 3$). The statistical significance of the measurements was calculated using an unpaired, two-tailed t-test (P-values listed in the figure; ns, non-significant).

NPF dimerization (GST-WWCA) partially restores the activity of this mutant, presumably because one arm of the dimer binds to Arp2-ArpC1, whereas the other arm remains in close proximity and can deliver actin to Arp3. If all Cort$_{1-76}$ did were to compete with NPF binding to Arp3, it would have the same effects as the Arp3 mutant, i.e. decrease polymerization, which is another argument against the displacement model. That Cort$_{1-76}$ increases polymerization (Fig. 4a) suggests that sequences N- and C-terminal to the Arp3-binding motif in inactive Arp2/3 complex promote binding to F-actin, either directly or indirectly by stabilizing the short-pitch conformation, or both.

At the branch junction, the conformation of Arp2/3 complex changes dramatically as a result of a rotation of two blocks of subunits containing Arp2 (ArpC1, ArpC4, ArpC5, and the C-terminal helix of ArpC2) and Arp3 (ArpC3, and the globular domain of ArpC2) and flattening of the Arps[8–10]. Additionally, at the branch junction Arp2/3

complex contacts five subunits of the mother filament and two subunits of the daughter filament. Cort$_{1-76}$ may bind with higher affinity to the active conformation of Arp2/3 complex at the branch junction, establishing new contacts through sequences N- and C-terminal to its Arp3-binding motif in the inactive complex. Additionally, Cort$_{1-76}$ may participate in interactions with subunits of either the mother or daughter filaments. Indeed, while previous studies have assumed that cortactin binds the mother filament[19,23,25–27,29,37], a recent study shows that cortactin stabilizes SPIN90-Arp2/3 complex at the pointed end of F-actin, presumably by binding the daughter filament[38]. Most of the F-actin-binding region of cortactin is intrinsically disordered and could potentially bind either the mother or daughter filaments. The results obtained here cannot distinguish between these two modes of binding and are compatible with both. Thus, rather than by accelerating NPF release[25,29,37], we propose that cortactin's synergistic effect during

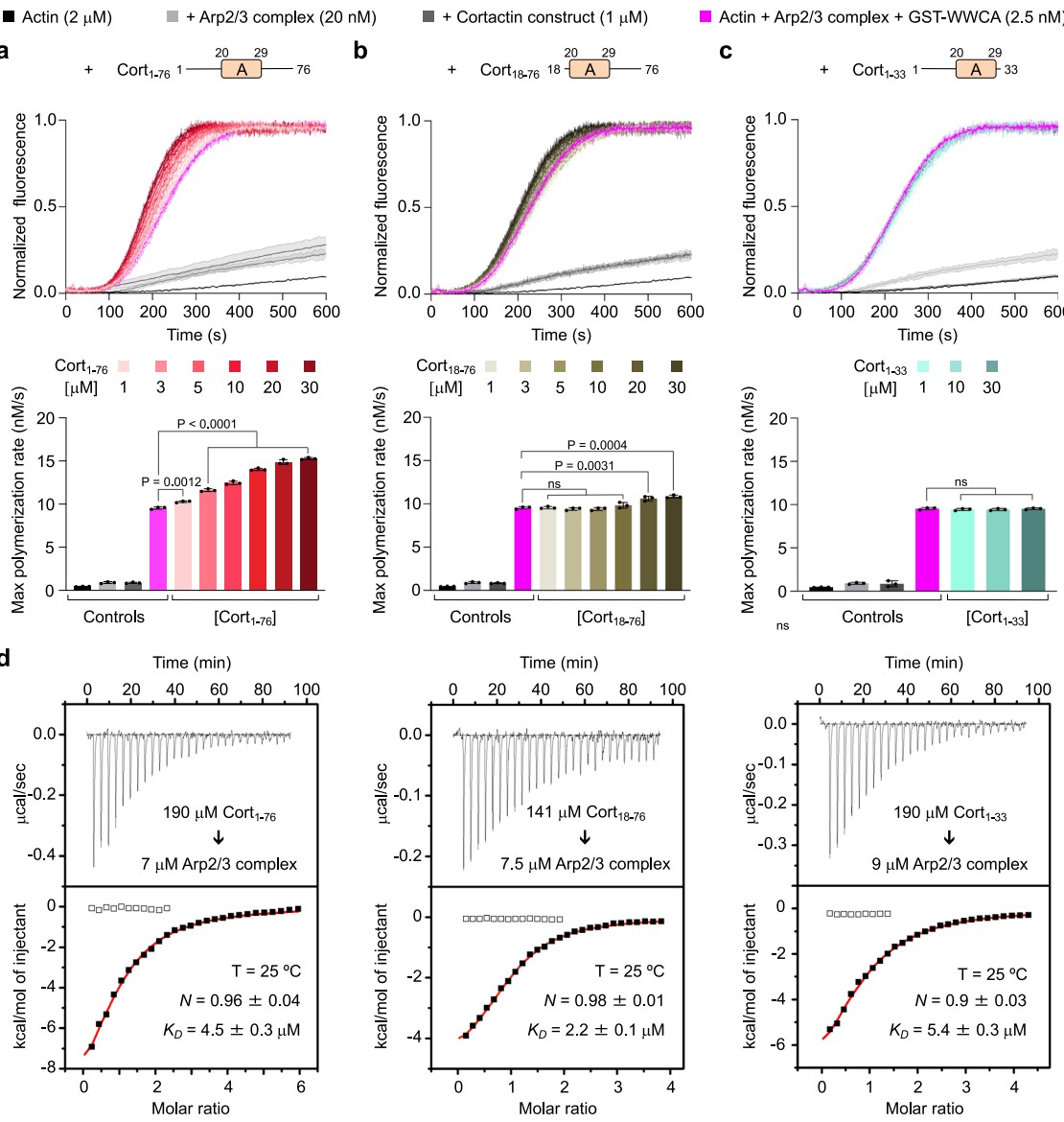

**Fig. 4 | Arp2/3 complex activation by N-terminal cortactin constructs with or without dimeric N-WASP WCA. a–c** Top, Time-course of actin polymerization under the following conditions: actin alone (black), actin + Arp2/3 complex (light gray), actin + Arp2/3 complex + indicated cortactin constructs (dark gray), actin + Arp2/3 complex + GST-WWCA (magenta), actin + Arp2/3 complex + GST-WWCA + indicated cortactin constructs at increasing concentrations (darker shades represent higher concentrations as indicated). Data are shown as the average curve from three independent experiments with s.d. error bars in lighter color. Bottom, Maximum polymerization rates calculated from the polymerization curves shown on the top ($n = 3$). Source Data are provided as a Source Data file. The statistical significance of the measurements was calculated using an unpaired, two-tailed t-test (P-values listed in the figure; ns, non-significant). **d** ITC titrations of cortactin constructs Cort$_{1-76}$ (left), Cort$_{18-76}$ (middle), and Cort$_{1-33}$ (right) into ATP-Arp2/3 complex. The experimental conditions, including temperature (T), protein concentration, and fitting parameters (stoichiometry, N and dissociation constant, $K_D$) are indicated.

Arp2/3 complex activation results from stabilization of the short-pitch conformation and recruitment of Arp2/3 complex, primed by interaction with actin-bound NPFs, to F-actin (Fig. 5).

Our model raises questions about the designation of cortactin as a class-II NPF[45], which can be confusing and should probably be abandoned. Both functionally and biochemically, cortactin differs fundamentally from conventional WASP-family NPFs whose primary role is to activate Arp2/3 complex in a regulated and localized manner[3]. Biochemically, cortactin does not promote the short-pitch transition the way classical NPFs do (Fig. 2). Functionally, Arp2/3 complex activation and branched network formation at sites typically associated with cortactin, such as lamellipodia, endocytic sites, and the comet tails of pathogens, is observed in cortactin-deficient cells[27,46], and cortactin-deficient mice are viable and fertile[47]. Thus, cortactin does not function as a classical NPFs (i.e. it is not required for Arp2/3

complex activation); at best, it synergizes with NPFs for more efficient branch network assembly and stabilization[14].

In summary, we propose that cortactin contributes to Arp2/3 complex coactivation with NPFs in two ways, by consolidating the interaction of Arp2/3 complex with F-actin (mother or daughter filament) and by stabilizing the short-pitch conformation at the branch junction, which are both byproducts of cortactin's core function in branch stabilization[23,26,27] (Fig. 5).

## Methods
### Proteins
The cDNA-encoding mouse cortactin (UniProt: Q60598) was obtained from American Type Culture Collection (ATCC). Constructs Cort$_{1-33}$, Cort$_{18-76}$ Cort$_{1-76}$, Cort$_{1-227}$, Cort$_{1-79/154-401}$, and Cort$_{1-401}$ were amplified by PCR and cloned between the BamHI and EcoRI sites of vector pMAL-

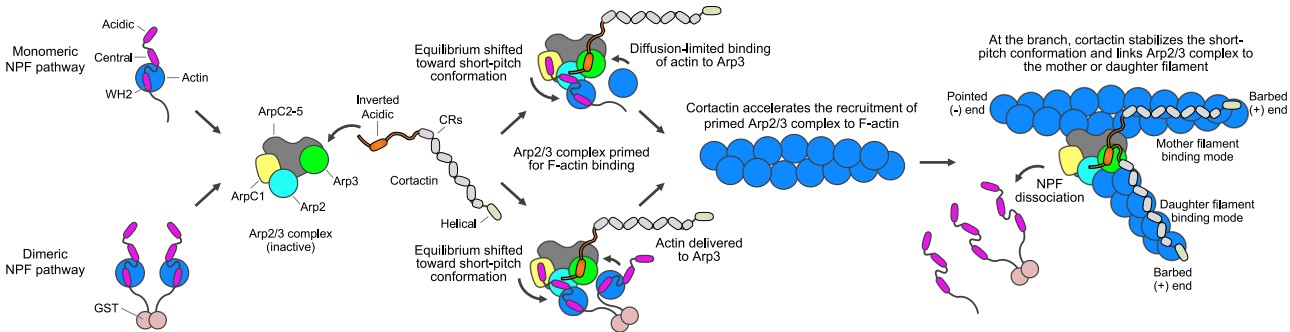

**Fig. 5 | Mechanism of cortactin coactivation of Arp2/3 complex with NPFs and branch stabilization.** Actin-loaded monomeric (top) or dimeric (bottom) NPFs bind inactive Arp2/3 complex through the high-affinity site 1 on Arp2-ArpC1. Cortactin competes with NPF binding to site 2 on Arp3, but not to site 1. NPF binding to Arp2-ArpC1 shifts the equilibrium of Arp2/3 complex toward the active, short-pitch conformation and delivers actin at the barbed end of Arp2, whereas cortactin recruits this "primed" complex to the mother filament. In the case of monomeric NPFs, the binding of actin at the barbed end of Arp3 is diffusion limited. For dimeric NPFs, however, one arm of the dimer is bound to Arp2-ArpC1, whereas the other

arm is close enough to deliver actin to Arp3. Binding of the primed complex to the side of the mother filament shifts the equilibrium further in the direction of the active, short-pitch conformation. The actin subunits that begin adding at the barbed end of the Arps compete with NPF binding to sites 1 and 2, causing NPF release. Upon NPF release, the short-pitch conformation is fully stabilized through interactions of Arp2/3 complex with actin subunits of the mother and daughter filaments. Cortactin remains bound at the branch junction, interacting with either the mother or daughter filament, which increases the stability of the branch.

c6T (NEB) comprising an MBP domain for affinity purification and a TEV protease site for affinity-tag removal. To obtain $Cort_{1-79/154-401}$, the coding sequence for residues 77–79 (GGACCCAAG) was used to add complementary overhangs onto fragments $Cort_{1-76}$ and $Cort_{154-401}$ for ligation (primers listed in Supplementary Table 1). Proteins were expressed in ArticExpress (DE3) cells (Agilent, catalog number 230192) and grown in Terrific Broth (TB) medium at 37 °C to an optical density at 600 nm ($OD_{600}$) of ~1.5 to 2. Expression was induced with 0.5 mM isopropylthio-$\beta$-galactoside (IPTG) and carried out for 20 h at 10 °C. Cells were harvested by centrifugation, resuspended in 50 mM Tris-HCl pH 8.0, 500 mM NaCl, and 100 µM phenylmethylsulfonyl fluoride (PMSF) and lysed on a microfluidizer (Microfluidics). Clarified lysates were loaded onto an amylose column (NEB) pre-equilibrated with 20 mM HEPES pH 7.5, 200 mM NaCl, extensively washed with the same buffer, and eluted with 10 mM maltose. TEV cleavage was carried out at 4 °C overnight, during dialysis against 20 mM Tris-HCl pH 7.5, 200 mM NaCl, 10 mM imidazole. Free MBP and TEV protease, both containing a His-tag, were removed on a Ni-NTA affinity column (QIAGEN). Proteins were additionally purified by size exclusion on a SD75HL 16/60 column (GE Healthcare) and concentrated using a Vivaspin Turbo 3 kDa concentrator (Sartorius). Protein concentrations were measured using theoretical extinction coefficients at 280 nm: $Cort_{1-33}$ (11,000 $M^{-1}cm^{-1}$), $Cort_{18-76}$ (11,000 $M^{-1}cm^{-1}$), $Cort_{1-76}$ (16,500 $M^{-1}cm^{-1}$), $Cort_{1-227}$ (29,910 $M^{-1}cm^{-1}$), $Cort_{1-79/154-401}$ (39,880 $M^{-1}cm^{-1}$), and $Cort_{1-401}$ (44,350 $M^{-1}cm^{-1}$).

Mouse N-WASP construct WCA was cloned into vector pTYB12 (NEB) and construct WWCA was cloned into the GST-coding vector pGEX-6p-1 (GE Healthcare). Proteins were expressed in ArticExpress (DE3) cells and purified using chitin- and glutathione-affinity columns, respectively[6]. Human Arp2/3 complex, used in crosslinking assays, featuring mutations in Arp2 (L199C) and Arp3 (L117C) was cloned using the biGBac system[48] and expressed in Sf9 cells (ATCC, catalog number CRL-1711)[6]. Additional mutations in Arp3 (C189A, C307A, C408A) were introduced to avoid non-specific crosslinking. Skeletal α-actin and bovine brain Arp2/3 complex were purified as described[49].

## Cryo-EM sample preparation and data collection
Bovine Arp2/3 complex (9.0 µM) was mixed with $Cort_{1-76}$ (45 µM) in ATP-supplemented KMEI buffer (10 mM imidazole pH 7.0, 50 mM KCl, 2 mM $MgCl_2$, 1 mM EGTA, and 0.2 mM ATP). Cryo-EM grids were prepared with the addition of 0.0025% NP-40, which improves particle distribution[6]. Samples (2 µL) were applied onto glow-discharged

(1 minute, easiGlow, Pelco) R1.2/1.3 200-mesh Quantifoil holey carbon grids (Electron Microscopy Sciences). The grids were blotted for 2 seconds with Whatman 41 filter paper and flash-frozen by plunging into liquid ethane using a Leica EM CPC manual plunger. For $Cort_{1-401}$-bound F-actin, rabbit skeletal α-actin (5 µM) was polymerized for 30 min in F-actin buffer (10 mM Tris pH 7.5, 1 mM $MgCl_2$, 50 mM KCl, 1 mM EGTA, 0.1 mM NaN3, and 0.2 mM ATP) and incubated with $Cort_{1-401}$ (25 µM) for another 30 min. Samples (3 µM) were applied onto glow-discharged (1 minute, easiGlow, Pelco) R1.2/1.3 300-mesh Quantifoil holey carbon grids (Electron Microscopy Sciences). The grids were blotted for 2.5 seconds at force 5 with Whatman 41 filter paper and flash-frozen by plunging into liquid ethane using a Vitrobot Mark IV.

Cryo-EM datasets were collected automatically using the EPU software (ThermoFisher Scientific) on a FEI Titan Krios transmission electron microscope operating at 300 kV and equipped with a K3 (Gatan) direct electron detector with an energy quantum filter (Gatan). For $Cort_{1-76}$-Arp2/3 complex, images (4106) were collected at a defocus range of −1.0 to −3.0 µm and a nominal magnification of 105,000x in super-resolution mode, resulting in a nominal pixel size of 0.42 Å. For $Cort_{1-401}$-bound F-actin, images (227) were collected at a defocus range of −0.8 to −2.5 µm and a nominal magnification of 64,000x in super-resolution mode, resulting in a nominal pixel size of 0.68 Å (Supplementary Fig. 6c).

## Cryo-EM data processing and model building
The $Cort_{1-76}$-Arp2/3 complex dataset was binned during motion-correction using MotionCor2[50] and imported into cryoSPARC (version v3.3.1)[51] for contrast transfer function (CTF) correction with CTFFIND4[52]. Micrographs were sorted based on CTF correction, resulting in a subset of 3,791 micrographs. CryoSPARC blob picking, followed by 2D classification produced 2D classes for reference-based template picking and the first ab initio model. Two rounds of heterogenous refinement and one round of non-uniform refinement resulted in a 3.69-Å reconstruction, used for iterative rounds of heterogenous refinement with the reference-picked particles. Template picking in cryoSPARC resulted in a total of 1,461,919 particles, and iterative rounds of 3D heterogenous refinement and 3D nonuniform refinement produced a 3.40-Å reconstruction. The particles used in this reconstruction (241,506) were transferred into Relion for further processing. Iterative rounds of CTF refinement (including beam tilt, per particle defocus, trefoil, and 4th order aberrations) and 3D refinement yielded

a 2.89 Å resolution map (half-map FSC = 0.143) after post processing with deepEMhancer[53]. Particles in the final reconstruction are uniformly distributed as determined with cryoEF[54], and the efficiency of the map ($E_{od}$) was 0.71. 3DFSC (https://3dfsc.salk.edu/) was used to calculate the 3D-FSC of the final map, which shows high sphericity (0.976 out of 1) and a global resolution of 2.96 Å (Supplementary Fig. 2b–d).

For Cort$_{1-401}$-bound F-actin, micrographs were imported into cryoSPARC and binned during patch motion correction. After patch CTF estimation, particles were picked using filament tracer and curated through two rounds of reference-free 2D classification. Helical refinement was used to reconstruct a total of 264,652 particles, followed by global and local CTF refinement. The final helical refinement yielded a 2.77 Å map.

The 2-Å resolution crystal structure of Arp2/3 complex alone (PDB code: 1K8K) was used as the starting model for refinement of the Cort$_{1-76}$-Arp2/3 complex. Multiple rounds of refinement using the program Phenix[55] and model building in Coot[56] led to a final model with good stereochemical parameters and good map-to-model correlation (Table 1 and Supplementary Fig. 2e). Figures were prepared using the programs ChimeraX[57] and PyMOL (Schrödinger, LLC).

### F-actin cosedimentation

Actin (40 μM) was polymerized in ATP-supplemented KMEI buffer. For cosedimention with cortactin, increasing concentrations of cortactin constructs were mixed with a constant concentration of F-actin (4 μM) (Supplementary Fig. 5). For cosedimention with Arp2/3 complex, 1.5 μM Arp2/3 complex with or without cortactin constructs (3 μM) was mixed with increasing concentrations of F-actin (Supplementary Fig. 7). Samples (100 μL) were incubated for 1 h at RT and centrifuged for 30 min at 278,800 g using a TLA-100 rotor. Supernatants were mixed with 25 μL 4xSDS loading dye. Pellets were washed with 100 μL KMEI buffer, resuspended in 100 μL KMEI buffer, and mixed with 25 μL 4xSDS loading dye. Samples were analyzed by SDS-PAGE and band intensities were quantified using densitometry in Image Lab 6.1 (Bio-Rad).

### Actin polymerization

Polymerization assays were carried out on a Cary Eclipse fluorescence spectrophotometer (Varian). Mg-ATP-actin (2 μM, 6% pyrene-labeled) was mixed with Arp2/3 complex (20 nM), and the indicated concentrations (Figs. 3, 4) of cortactin constructs, N-WASP WCA or GST-WWCA in F-buffer. The maximum polymerization rate was calculated using the equation S' = (S x M$_t$)/(f$_{max}$ − f$_{min}$), where S' is the apparent slope in μM s$^{-1}$, S is the maximum slope of the raw trace, M$_t$ is the concentration of polymerizable monomers, f$_{max}$ is the maximum fluorescence intensity at plateau, and f$_{min}$ is the fluorescence intensity at the start of the reaction. The statistical significance of the measurements was determined with the program Prism v7.0 using an unpaired two-way Student's t-test (P-values are reported in the figures).

### Short-pitch crosslinking assay

Human Arp2/3 complex (crosslinking mutant, 0.25 μM) was incubated with 1.5 μM WCA, 5 μM Cort$_{1-76}$, or 1.5 μM WCA + 5 μM Cort$_{1-76}$ for 30 min at RT in ATP-supplemented KMEI buffer. Freshly prepared BMOE (ThermoFisher Scientific) in dimethyl sulfoxide (DMSO) was added to a final concentration of 16 μM at 21 °C. Reactions were quenched at the indicated time points (Fig. 2) with the addition of an equal volume of 2x SDS-loading buffer (LI-COR Biosciences) supplemented with 100 mM β-mercaptoethanol. Samples were loaded onto 12% SDS-PAGE gels, transferred onto polyvinylidene fluoride (PVDF) membranes (Bio-Rad) and immunoblotted with anti-Arp3 antibody (1:5000 dilution, Santa Cruz Biotechnology, sc-48344). Membranes were imaged using a G:BOX scanner (Syngene). Densitometric analysis

was performed using Image Lab 6.1 (Bio-Rad). Mean and SD values were calculated from three independent experiments.

### Reporting summary

Further information on research design is available in the Nature Portfolio Reporting Summary linked to this article.

## Data availability

The cryo-EM map and atomic model generated for this work is deposited in the Electron Microscopy Data Bank and Protein Data Bank with accession codes EMD-41135 and 8TAH, respectively. The PDB accession codes for existing models referenced here are 6UHC and 7JPN. Uniprot database was used to obtain the sequence for mouse cortactin Q60598. Uncropped western blots and gels used in quantifications for this study are shown in Supplementary Information. The biochemical data generated in this study are provided in the Source Data file. Source data are provided with this paper.

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

## Acknowledgements

Supported by National Institutes of Health grants R01 GM073791 to R.D. and F31 GM148048 to F.E.F. Data collection was performed at the Electron Microscopy Resource Lab (EMRL) and The Beckman Center for

Cryo-Electron Microscopy, University of Pennsylvania (Research Resource Identifier SCR_022375). We thank S. Steimle for assistance with data collection. Computational recourses at the University of Pennsylvania were supported by NIH instrumentation grant S10OD023592.

## Author contributions

F.E.F. and R.D. designed the research. F.E.F. obtained the cortactin proteins, performed the pyrene-actin polymerization assays and obtained the cryo-EM structure of Arp2/3 complex with bound cortactin. M.B. obtained human Arp2/3 complex, performed the crosslinking experiments, and ran cosedimentation assays. G.R. purified actin and bovine Arp2/3 complex and performed the ITC experiments. P.J.C. participated in the execution and analysis of the cosedimentation experiments and performed the cryo-EM analysis of actin filament bundles with cortactin. T.v.E. participated in the analysis of the cryo-EM data. R.D. wrote the paper. F.E.F., R.D., and M.B. prepared the figures.

## Competing interests

The authors declare no competing interests.
