## [Peer Review File · Nature Communications]

REVIEWER COMMENTS

Reviewer #1 (Remarks to the Author):

Synergetic activation of ARP2/3 complex by NPFs and cortactin is a crucial element of actin networks regulation in motile cells. Fregoso et al use high resolution structure of the inactive ARP2/3 complexed with the acidic domain of cortactin along with biochemical and kinetic approaches to propose a model of co-activation of ARP2/3 by WASP and cortactin. The work is of a high quality and should be interesting to a broad scientific community. I have the following questions and suggestions:

1. In the absence of ARP2/3 the interaction of Cort1-401 with F-actin lacks specificity and Cort1-401 is not visualized on F-actin by cryo-EM despite bundling effect (Extended Data Fig. 7). How these observations are related to the previously published structure of F-actin decorated with actin-binding repeating domain of cortactin (Pant et al., JMB 2006)? The experimental conditions (lacking in the manuscript) should be compared with the previously published data and discussed.
2. Cort1-76 increases polymerization (Fig. 4a) which suggests that residues flanking the APR3 binding motif (invisible in the cryo-EM map (Fig. 1b)) promote binding of ARP2/3 to the mother filament either:
 - 1) directly, but this is inconsistent with the lack of recruitment of APR2/3 to F-actin by Cort1-76 (Fig. 3a),
 - 2) or by indirect stabilization of the activated (e.g. short-pitch) conformation of ARP2/3 which is inconsistent with the cross-linking data (Fig. 2).

These inconsistencies should be carefully addressed in the manuscript. Maybe the model of the activated complex (Extended Data Fig. 3) may be used to support the idea of short-pitch stabilization by Cort1-76 (e.g. obvious contacts between cortactin and Arp2 or cortactin and the mother filament)? Is it possible to do cryo-EM structure of the cross-linked APR2/3 complex with Cort1-76 to directly show the stabilization of the activated ARP2/3 by the N-terminus of cortactin?

Minor points:

1. Fig. 3: What is the symbol used for concentrations (∞ M)? Y axis numbers in b, c, d, and e are clipped.
2. Fig. 4: X axis numbers in a are clipped.

Reviewer #2 (Remarks to the Author):

This is a very interesting manuscript that aims at defining how precisely the Arp2/3 complex and actin filament binding protein cortactin contributes to activation of Arp2/3 complex-mediated actin assembly and F-actin branching. It focusses in particular on the precise consequence of binding of the N-terminus of cortactin to the Arp2/3 complex subunit Arp3, and how the inverted acidic domain (residues 20-29)

plus surrounding sequence stretches contribute to the efficiency of Arp2/3 complex activation, in particular in synergy with classical NPF-C-termini, such as WCA or WWCA of N-WASP. The authors start off with a 2.89-Å cryo-EM structure of cortactin's N-terminal 76 amino acid residues bound to soluble, inactive Arp2/3 complex. Biochemical assays, such as an elegant short-pitch crosslinking assay or classical pyrene-actin assembly assays complement the story to explore how precisely this interaction promotes Arp2/3-dependent actin assembly. Doing this, the authors distinguish between three models of synergistic Arp2/3 activation, (i) direct-activation, which is discarded based on the authors' results, interestingly, (ii) the so called recruitment model, which can explain most of the observations, at least in part, and (iii) the displacement model, which again appears largely incompatible with the authors' data and conclusions derived from them.

I believe this is a very solid study, which deserves to be rapidly published, and I largely agree with all the interpretations and conclusions drawn by the authors. I do have a few thoughts and suggestions though for potential, experimental clarification or improvement of the text or some, ambiguous statements, but these are very minor and should not lead to extensive delays of final acceptance and publication.

Specific comments:

1) In line 42 in the intro, the authors state that cortactin is found in branched actin networks throughout the cell, which is correct, but then mention "cell adhesions" and "invadopodia" amongst other subcellular structures. Invadopodia are cortactin-containing, Arp2/3-dependent structures and thus correct, of course, as well as the related, hematopoietic structures known as podosomes (which are not mentioned), but the most prominent type of "cell adhesions" are focal adhesions or focal contacts, which definitely lack Arp2/3 complex and thus cortactin, so I feel the statement should be amended as follows. The authors could either remove the term "cell adhesions" from the list, or replace it by 'podosomes in hematopoietic cells', or they could replace "invadopodia" by 'invadosomes', which is a common term including both podosomes and invadopodia.

2) In lines 130-2 (and in line 233 in the discussion), the authors discard the direct-activation model of Arp2/3 complex by cortactin. In lines 130-2, it is stated that "unlike class I NPFs, cortactin (as class II-NPF) does NOT shift the equilibrium of Arp2/3 complex toward the short-pitch conformation in solution, thereby ruling out the direct-activation model." This is quite important and exciting indeed.

So I agree with this conclusion, but I feel it could be emphasized even further, because at least part of the community still considers cortactin an important activator of Arp2/3 complex-dependent actin assembly, although the current data suggest (at least in my view) that it cannot execute the function in the absence of a class I NPF. Instead, it can only tune or optimize the process (see also Schnoor et al., 2018, PMID: 29162307 for a discussion of this issue). So in this context, I think the authors could also cite research here from genetic knockouts in cells, which emphasize that cortactin-containing structures are still formed and Arp2/3 complex accumulation (and hence activation) still occurs if cortactin is completely missing. This is true for structures as prominent as lamellipodia and sites of endocytosis (see Lai et al., 2009, PMID: 19458196) or Vaccinia virus actin tails (Abella et al., 2016, PMID: 26655834 -

already cited by the authors in another context). Finally, in spite of certain, specific phenotypes that are emerging over the years, cortactin-deficient mice are viable and fertile (Schnoor et al., 2011, PMID: 21788407). So in other words, genetic and cell biological data - even from higher animals - suggest that lack of cortactin does not likely lead to the complete loss of a specific, cortactin-dependent actin structure in vivo, which perfectly fits the conclusion here that cortactin alone is incapable Arp2/3 complex activation - so why not mention that?

3) In lines 175-7, the authors argue that “Cort1-76, which cannot recruit Arp2/3 to F-actin on its own, also synergized with monomeric WCA during Arp2/3 complex activation, albeit to a lesser extent compared to other cortactin constructs” (the latter of which harboring F-actin binding domains). However, when considering the max polymerization rate at least, the extent of this synergetic effect is minute, in my view, at least as compared to cortactin harboring the full F-actin repeats and helical domain (Cort1-401). So in this respect, I feel this conclusion is too strong, as it could also be interpreted as the N-terminus alone, at least if employed with monomeric WCA, is not capable of significantly improving Arp2/3-dependent actin assembly further! So I wondered if this argument should be weakened. In any case, this brings me to the last point.

4) I would agree again with the authors' interpretation that Cort1-76 can indeed enhance the polymerization rate in the experiment employing dimeric, GST-WWCA, and in a concentration-dependent fashion, which can be found in the last section of the Results, where the authors examine the displacement model of Arp2/3 complex activation by cortactin (Fig. 4a). So here, there is a measureable increase in efficiency of Arp2/3-dependent actin assembly, at least in these pyrene-assay conditions, and, interestingly, this requires additional sequence stretches N- and C-terminal to the core Arp3-binding unit (no effect for Cort18-76 or 1-33, Fig. 4b and 4c). However, when I compare Fig. 3e with 4a, the dimerization (through GST) is not the only difference between these two experiments. Although it is possible that I am missing something here, the experiments in Fig. 4 employ the C-terminus of N-WASP with both of its WH2-domains (i.e. WWCA fused to GST), whereas Fig. 3e uses monomeric WCA. So can the authors exclude that the presence of the second, actin binding WH2-domain (as it occurs in the native N-WASP sequence) could also lead to slightly differential results? This could be confusing, at least for the naïve reader. So since it is always tricky in such complex experimental settings to change two parameters at the same time (of course the concentrations of the two NPF-versions are also different, but this is trivial), I would recommend specifying in the text if an impact on expt. outcome of using WCA versus WWCA can be excluded, and if yes, why this is the case. Alternatively, and even better, the authors could repeat the experiment with GST-WCA (dimerized WCA) and/or with WWCA (monomeric WWCA), at least for Cort1-76, to see whether the presence of the second WH2-domain makes any difference. This would be excellent.

5) In lines 252/3, the authors refer to a previous study on the displacement model of synergistic Arp2/3 activation by cortactin, and state that these previous authors used slightly different constructs (Helgeson et al., 2014, PMID: 25160634). It would be great if this could be specified. Specifically, do the authors want to infer that the fact that the constructs were slightly different could make a difference in

expt. outcome and conclusion, or do the authors see additional, potential reasons for the diverging interpretation? Being specific about this point would prevent the reader having 'to read between the lines'.

Response to Reviewers (blue text)

We would like to thank the reviewers for their meticulous reading of the manuscript and excellent suggestions, which in our opinion improve the manuscript. We changed the model in Fig. 5 to reflect that cortactin binding to the mother or daughter filament have both been proposed. Our data is compatible with both modes of binding.

Reviewer 1

Synergetic activation of ARP2/3 complex by NPFs and cortactin is a crucial element of actin networks regulation in motile cells. Fregoso et al use high resolution structure of the inactive ARP2/3 complexed with the acidic domain of cortactin along with biochemical and kinetic approaches to propose a model of co-activation of ARP2/3 by WASP and cortactin. The work is of a high quality and should be interesting to a broad scientific community. I have the following questions and suggestions:

We appreciate the very positive comments of the reviewer and address the questions and comments made in our response below, as well as in the revised text, which we believe has enhanced the manuscript.

1. In the absence of ARP2/3 the interaction of Cort1-401 with F-actin lacks specificity and Cort1-401 is not visualized on F-actin by cryo-EM despite bundling effect (Extended Data Fig. 7). How these observations are related to the previously published structure of F-actin decorated with actin-binding repeating domain of cortactin (Pant et al., JMB 2006)? The experimental conditions (lacking in the manuscript) should be compared with the previously published data and discussed.

As suggested, we added the sample preparation conditions for helical reconstruction in the Methods (Lines 359-360). Concerning Pan et al., 2006 - the resolution of this study was low (23Å) and it used negative staining EM, known to produce artifacts (PMID: 29443097), which most likely explains the different results (as opposed to differences in sample preparation). We now mention this point in the manuscript (Lines 147-150)

2. Cort1-76 increases polymerization (Fig. 4a) which suggests that residues flanking the APR3 binding motif (invisible in the cryo-EM map (Fig. 1b)) promote binding of ARP2/3 to the mother filament either: 1) directly, but this is inconsistent with the lack of recruitment of APR2/3 to F-actin by Cort1-76 (Fig. 3a), or 2) by indirect stabilization of the activated (e.g. short-pitch) conformation of ARP2/3 which is inconsistent with the cross-linking data (Fig. 2). These inconsistencies should be carefully addressed in the manuscript. Maybe the model of the activated complex (Extended Data Fig. 3) may be used to support the idea of short-pitch stabilization by Cort1-76 (e.g. obvious contacts between cortactin and Arp2 or cortactin and the mother filament)? Is it possible to do cryo-EM structure of the cross-linked APR2/3 complex with Cort1-76 to directly show the stabilization of the activated ARP2/3 by the N-terminus of cortactin?

We now added the specific questions asked by the reviewer to our own questions in the Discussion (Lines 249-250) and reworded a section of the Discussion to specifically answer the questions asked by the reviewer (Lines 287-303). After careful consideration we removed the model of old Extended Data Fig. 3, because there are two proposed modes of cortactin binding at the branch, to the mother or the daughter filaments, and our data cannot rule out between these two models. This is explained in the same paragraph of the Discussion.

Regarding the possibility of obtaining a structure of BMOE-crosslinked Arp2/3 complex – currently, this is not a viable approach for us because we do not have a reliable way to separate the crosslinked and un-crosslinked complexes. Further, for mammalian Arp2/3 complex the reaction is not easily scaled up, such as to obtain enough sample for structural studies. Finally, the crosslinked complex is unlikely to faithfully represent the native short-pitch conformation at the branch junction, since other factors contribute to reaching this conformation (please, see our recent work: van Eeuwen *PNAS* 2023).

Minor points:

1. Fig. 3: What is the symbol used for concentrations (∞ M)? Y axis numbers in b, c, d, and e are clipped.
2. Fig. 4: X axis numbers in a are clipped.

We thank the reviewer for noticing these issues, resulting from image reformatting. These have been fixed.

Reviewer 2

This is a very interesting manuscript that aims at defining how precisely the Arp2/3 complex and actin filament binding protein cortactin contributes to activation of Arp2/3 complex-mediated actin assembly and F-actin branching. It focusses in particular on the precise consequence of binding of the N-terminus of cortactin to the Arp2/3 complex subunit Arp3, and how the inverted acidic domain (residues 20-29) plus surrounding sequence stretches contribute to the efficiency of Arp2/3 complex activation, in particular in synergy with classical NPF-C-termini, such as WCA or WWCA of N-WASP. The authors start off with a 2.89-Å cryo-EM structure of cortactin's N-terminal 76 amino acid residues bound to soluble, inactive Arp2/3 complex. Biochemical assays, such as an elegant short-pitch crosslinking assay or classical pyrene-actin assembly assays complement the story to explore how precisely this interaction promotes Arp2/3-dependent actin assembly. Doing this, the authors distinguish between three models of synergistic Arp2/3 activation, (i) direct activation, which is discarded based on the authors' results, interestingly, (ii) the so-called recruitment model, which can explain most of the observations, at least in part, and (iii) the displacement model, which again appears largely incompatible with the authors' data and conclusions derived from them.

I believe this is a very solid study, which deserves to be rapidly published, and I largely agree with all the interpretations and conclusions drawn by the authors. I do have a few thoughts and suggestions though for potential, experimental clarification, or improvement of the text or some, ambiguous statements, but these are very minor and should not lead to extensive delays of final acceptance and publication.

We would like to thank the reviewer for the very positive and thoughtful comments. Below, we address the reviewer's comments, which clearly improve the manuscript. Specifically, the comments regarding the value of this work for the interpretation of cellular and animal work on cortactin are, in our opinion, a great addition to the manuscript.

Specific comments:

- 1) In line 42 in the intro, the authors state that cortactin is found in branched actin networks throughout the cell, which is correct, but then mention "cell adhesions" and "invadopodia"

amongst other subcellular structures. Invadopodia are cortactin-containing, Arp2/3-dependent structures and thus correct, of course, as well as the related, hematopoietic structures known as podosomes (which are not mentioned), but the most prominent type of “cell adhesions” are focal adhesions or focal contacts, which definitely lack Arp2/3 complex and thus cortactin, so I feel the statement should be amended as follows. The authors could either remove the term “cell adhesions” from the list, or replace it by ‘podosomes in hematopoietic cells’, or they could replace “invadopodia” by ‘invadosomes’, which is a common term including both podosomes and invadopodia.

This is very helpful. As suggested, we removed “cell adhesions”, and replaced “invadopodia” with “invadosomes” (Line 43).

2) In lines 130-2 (and in line 233 in the discussion), the authors discard the direct-activation model of Arp2/3 complex by cortactin. In lines 130-2, it is stated that “unlike class I NPFs, cortactin (a class II NPF) does NOT shift the equilibrium of Arp2/3 complex toward the short-pitch conformation in solution, thereby ruling out the direct-activation model.” This is quite important and exciting indeed. So, I agree with this conclusion, but I feel it could be emphasized even further, because at least part of the community still considers cortactin an important activator of Arp2/3 complex-dependent actin assembly, although the current data suggest (at least in my view) that it cannot execute the function in the absence of a class I NPF. Instead, it can only tune or optimize the process (see also Schnoor et al., 2018, PMID: 29162307 for a discussion of this issue). So, in this context, I think the authors could also cite research here from genetic knockouts in cells, which emphasize that cortactin-containing structures are still formed and Arp2/3 complex accumulation (and hence activation) still occurs if cortactin is completely missing. This is true for structures as prominent as lamellipodia and sites of endocytosis (see Lai et al., 2009, PMID: 19458196) or Vaccinia virus actin tails (Abella et al., 2016, PMID: 26655834 - already cited by the authors in another context). Finally, in spite of certain, specific phenotypes that are emerging over the years, cortactin-deficient mice are viable and fertile (Schnoor et al., 2011, PMID: 21788407). So, in other words, genetic and cell biological data - even from higher animals - suggest that lack of cortactin does not likely lead to the complete loss of a specific, cortactin-dependent actin structure in vivo, which perfectly fits the conclusion here that cortactin alone is incapable Arp2/3 complex activation - so why not mention that?

We are very grateful to the reviewer for these important suggestions. We added an entire new paragraph to the Discussion to develop these ideas (Lines 304-313). Again, a big thank you!

3) In lines 175-7, the authors argue that “Cort1-76, which cannot recruit Arp2/3 to F-actin on its own, also synergized with monomeric WCA during Arp2/3 complex activation, albeit to a lesser extent compared to other cortactin constructs” (the latter of which harboring F-actin binding domains). However, when considering the max polymerization rate at least, the extent of this synergetic effect is minute, in my view, at least as compared to cortactin harboring the full F-actin repeats and helical domain (Cort1-401). So in this respect, I feel this conclusion is too strong, as it could also be interpreted as the N-terminus alone, at least if employed with monomeric WCA, is not capable of significantly improving Arp2/3-dependent actin assembly further! So I wondered if this argument should be weakened. In any case, this brings me to the last point.

The indicated effect is nonetheless observable and statistically validated, so the sentence is correct (Fig. 3e). Yet, in agreement we the reviewer, we have further weakened the conclusion by adding “much”, i.e. “albeit to a MUCH lesser extent compared to other cortactin constructs”

(Line 176)

4) I would agree again with the authors' interpretation that Cort1-76 can indeed enhance the polymerization rate in the experiment employing dimeric, GST-WWCA, and in a concentration-dependent fashion, which can be found in the last section of the Results, where the authors examine the displacement model of Arp2/3 complex activation by cortactin (Fig. 4a). So here, there is a measurable increase in efficiency of Arp2/3-dependent actin assembly, at least in these pyrene-assay conditions, and, interestingly, this requires additional sequence stretches N- and C-terminal to the core Arp3-binding unit (no effect for Cort18-76 or 1-33, Fig. 4b and 4c). However, when I compare Fig. 3e with 4a, the dimerization (through GST) is not the only difference between these two experiments. Although it is possible that I am missing something here, the experiments in Fig. 4 employ the C-terminus of N-WASP with both of its WH2-domains (i.e. WWCA fused to GST), whereas Fig. 3e uses monomeric WCA. So can the authors exclude that the presence of the second, actin binding WH2-domain (as it occurs in the native N-WASP sequence) could also lead to slightly differential results? This could be confusing, at least for the naïve reader. So since it is always tricky in such complex experimental settings to change two parameters at the same time (of course the concentrations of the two NPF-versions are also different, but this is trivial), I would recommend specifying in the text if an impact on expt. outcome of using WCA versus WWCA can be excluded, and if yes, why this is the case. Alternatively, and even better, the authors could repeat the experiment with GST-WCA (dimerized WCA) and/or with WWCA (monomeric WWCA), at least for Cort1-76, to see whether the presence of the second WH2-domain makes any difference. This would be excellent.

There are several reasons why we used GST-WWCA, and not the single W domain construct. First, it is important to point out that, alone, WCA is a stronger activator of Arp2/3 complex than WWCA, both as monomeric (PMID: 32917641; 11747816; 25160634) or dimeric (PMID: 25160634) constructs. Second, GST-WWCA has been extensively used, including by us (PMID: 32917641) and others (PMID: 11747816; 23623350; 24015358; 18995840; 21676863; 25160634), to quote but a few examples. We have opted to use the same dimeric GST-WWCA construct others have used, which has two advantages: 1) the results are directly comparable to other studies, and 2) the reason (in retrospect) people have used two W domains is to add additional space between GST and WCA. Too short a linker eliminates the dimeric effect, likely because the two NPF-binding sites on Arp2/3 complex are not simultaneously reachable. Third, if we add GST to WCA using "any" longer linker, we will end up having a new untested construct with its own set of unknowns. Therefore, we believe it is better to use what most people have used and tested. We now clarify why we used this construct (Lines 183-186).

5) In lines 252/3, the authors refer to a previous study on the displacement model of synergistic Arp2/3 activation by cortactin, and state that these previous authors used slightly different constructs (Helgeson et al., 2014, PMID: 25160634). It would be great if this could be specified. Specifically, do the authors want to infer that the fact that the constructs were slightly different could make a difference in expt. outcome and conclusion, or do the authors see additional, potential reasons for the diverging interpretation? Being specific about this point would prevent the reader having 'to read between the lines'.

Good point. We added clarifications in Lines 258-261. Briefly, the results of PMID: 32917641 are similar to ours, but the interpretation is different. In the same and following paragraphs of the Discussion, we re-worded the text to clearly explain why we disagree with the interpretation of these authors, with our model being informed by recent advances, including this study and published work.